# Obeticholic acid treatment ameliorates the cardiac dysfunction in NASH mice

Szu-Yu Liu[1,2‡], Chia-Chang Huang[2,3‡], Ying-Ying Yang[1,2]*, Shiang-Fen Huang[2,4], Tzung-Yan Lee[5], Tzu-Hao Li[2,3,6], Ming-Chih Hou[2,4], Han-Chieh Lin[4]*

1 Department of Medical Education, Clinical Innovation Center, Medical Innovation and Research Office, Taipei Veterans General Hospital, Taipei, Taiwan, 2 Faculty of Medicine, School of Medicine, National Yang-Ming Chiao Tung University, Taipei, Taiwan, 3 Faculty of Medicine, Institute of Clinical Medicine, School of Medicine, National Yang-Ming Chiao Tung University, Taipei, Taiwan, 4 Taipei Veterans General Hospital, Taipei, Taiwan, 5 Graduate Institute of Traditional Chinese Medicine, Chang Guang Memorial Hospital, Linkou, Taiwan, 6 Division of Allergy, Immunology, and Rheumatology, Department of Internal Medicine, Shin Kong Wu Ho-Su Memorial Foundation, Taipei, Taiwan

‡ These authors share first authorship on this work.
* yangyy@vghtpe.gov.tw (Y-YY); hclin@vghtpe.gov.tw (H-CL)

**Data Availability Statement:** All relevant data are within the manuscript and its Supporting Information files.

**Funding:** This work was supported in part by MOST-110-2634-F-A49-005, MOST-109-2314-B-

## Abstract

### Background

Suppression of cardiac iinflammasome, which can be inhibited by Farnesoid X receptor (FXR) agonist, can ameliorate cardiac inflammation and fibrosis. Increased cardiac inflammasome decrease the abundance of regulatory T (Treg) cells and exacerbate cardiac dysfunction. Interaction between cardiomyocytes and Treg cells is involved in the development of nonalcoholic steatohepatitis (NASH)-related cardiac dysfunction.

### Aims

This study evaluates whether the FXR agonist obeticholic acid (OCA) treatment improves NASH-associated cardiac dysfunction.

### Methods

The *in vivo* and *in vitro* mechanisms and effects of two weeks of OCA treatment on inflammasome and Treg dysregulation-related cardiac dysfunction in NASH mice (NASH-OCA) at systemic, tissue and cellular levels were investigated.

### Results

The OCA treatment suppressed the serum and cardiac inflammasome levels, reduced the cardiac infiltrated CD3[+] T cells, increased the cardiac Treg-represented anti-inflammatory cytokines (IL-10/IL-10R) and improved cardiac inflammation, fibrosis and function [decreased left ventricle (LV) mass and increased fractional shortening (FS)] in NASH-OCA mice. The percentages of OCA-decreased cardiac fibrosis and OCA-increased FS were positively correlated with the percentage of OCA-increased levels of cardiac FXR and IL-10/IL-10R. In the Treg cells from NASH-OCA mice spleen, in comparison with the Treg cells of the NASH group, higher intracellular FXR but lower inflammasome levels, and more

010-032-MY3 and MOST-110-2511-H-A491-504-MY3 from the National Science Council (Y.Y.Y., H.C.L. and C.C.H.), 111EA-009, V111EA-010, V111C-018, V111C-038, VTA111-A-4-3 (Y.Y.Y. H.C.L. and C.C.H.) from the Taipei Veterans General Hospital, 111Q58501Y (Y.Y.Y.) from National Chiao Tung University, and PMN1110190 from Ministry of education (Y.Y.Y.). The funders had no role in study design, data collection and analysis, decision to publish, or preparation of the manuscript.

**Competing interests:** The authors have declared that no competing interests exist.

proliferative/active and less apoptotic cells were observed. Incubation of H9c2 cardiomyoblasts with Treg-NASHcm [supernatant of Treg from NASH mice as condition medium (cm)], increased inflammasome levels, decreased the proliferative/active cells, suppressed the intracellular FXR, and downregulated differentiation/contraction marker. The Treg-NASHcm-induced hypocontractility of H9c2 can be attenuated by co-incubation with OCA, and the OCA-related effects were abolished by siIL-10R pretreatment.

## Conclusions

Chronic FXR activation with OCA is a potential strategy for activating IL-10/IL-10R signalling, reversing cardiac regulatory T cell dysfunction, and improving inflammasome-mediated NASH-related cardiac dysfunction.

## Introduction

Non-alcoholic fatty liver disease (NAFLD), non-alcoholic steatohepatitis (NASH), fibrosis, and cirrhosis, are the most prevalent worldwide liver disease [1]. Blocking the activation of the inflammasome [nucleotide-binding oligomerization domain leucine-rich repeat and pyrin domain containing 3 (NLRP3)] and caspase-1 can reduce the progression of NASH [2]. Cardial dysfunction have been reported in NAFLD patients and are correlated with disease progression [3,4]. Left ventricle hypertrophy and ventricular diastolic dysfunction have been reported in patients with NAFLD [4]. Severe cardiac inflammation and fibrosis in diabetic rats are associated with the activation of cardiac inflammasome [5]. The NLRP3 inflammasome is abundantly expressed in cardiomyocytes [6,7]. Active caspase-1 mediates the activation of interleukin-1β (IL-1β). The NLRP3 inflammasome increases the secretion of active caspase-1 and IL-1β. Severe cardiac inflammation and fibrosis in diabetic rats are associated with the activation of cardiac NLRP3, caspase-1, and IL-1β [5]. In turn, NLRP3 inflammasome gene silencing improves diabetic cardiomyopathy by ameliorating cardiac inflammation and fibrosis [6]. It has thus been reported that the NLRP3 inflammasome promotes myocardial dysfunction through the production of IL-1β [8]. In contrast, IL-1 blockade reduces cardiac inflammation, ameliorates cardiac dysfunction and improves exercise capacity in mice and patients with heart failure [8,9].

Farnesoid X receptor (FXR) is a bile acid sensor expressed in the liver and heart, and its activation can limit the inflammatory response [10]. Farnesoid X receptor agonists inhibit the activation of the NLRP3 inflammasome and provide a potential therapeutic target for NLRP3-dependent inflammatory diseases [11]. The FXR agonist obeticholic acid (OCA) improves hepatic steatosis and inflammation by inhibiting hepatic and macrophage NLRP3 inflammasome activation [12–14]. In the NASH model, FXR agonist treatment decrease hepatic steatosis and inflammation [14]. Small heterodimer partner (SHP), a downstream target gene of FXR, has been shown to prevent NLRP3 formation [15].

Regulatory T (Treg) cells are a subset of T cells with anti-inflammatory effects and Treg cells in mice with hepatic steatosis are more susceptible to apoptosis [16]. Additionally, Treg cells in patients with chronic heart failure and dilated cardiomyopathy are susceptible to apoptosis and less suppressive activity when co-incubated with T cells [17,18]. Decreased circulatory and hepatic Treg cells have been reported in patients with NAFLD [19]. NLRP3 negatively regulates the abundance of Treg cells in the peripheral lymph nodes of mice [20].

Furthermore, there is a negative correlation between NLRP3 expression levels and the frequency of circulating Treg cells in patients with recurrent spontaneous abortions [21].

Serum and tissue interleukin-10 and its receptor IL-10R are markers of Treg cells activities. Interleukin-10 signalling in Treg cells is required to maintain anti-inflammatory effects [22]. IL-10 signalling via IL-10R has protective effects against pressure overload-induced cardiac hypertrophy in mice [23,24]. Systemic administration of IL-10 limit the progression of obesity-related cardiac inflammation and fibrosis [25]. In diabetic mice, FXR activation by OCA treatment improves myocardial dysfunction by inhibiting cardiac inflammation and fibrosis [26–28]. Accordingly, this study aimed to evaluate the mechanisms and effects of chronic treatment with the FXR agonist OCA on cardiac Treg cell dysfunction-related, inflammasome-mediated, and NASH-associated cardiac dysfunction in mice.

## Materials and methods

### Animals

This study was approved by the Animal Experiments Committee of Yang-Ming Chiao Tung University (1080404r) and was performed according to the "Guide for the Care and Use of Laboratory Animals" prepared by the National Academy of Science (USA) and the ARRIVE guidelines. At the end of the experiments, the mice were euthanised using two to three times the anaesthetic dose of Zoletil®. All efforts were made to minimise the number of animals necessary to produce reliable results, and suffering was reduced by administering anaesthetics (Zoletil® and xylocaine). Twelve-week-old C57BL/6 mice (Jackson Laboratories, Bar Harbour, ME) were fed with normal chow for 20 weeks (NC, Laboratory Autoclavable Rodent Diet 5010) and were designated the control (Ctrl) group, while the NASH group received a high fat and methionine and choline deficiency (MCD) diet (HFMCD) for 20 weeks. This diet was composed of 37% calories (Cal) from fat (corn oil), 24.5% Cal from protein (lactalbumin hydrolysate), 38.5% Cal from carbohydrates (dextrose), and vitamins and minerals (Dyets Inc., Bethlehem, PA, USA) as recommended [29–31]. The NASH-OCA group received obeticholic acid (OCA, a semi-synthetic bile acid), which acts as a FXR ligand and agonist, at a dose of 10 mg/kg/day for two weeks. The OCA or vehicle was administered using an alzet ® osmotic pump (DURECT, Cupertino, CA) or with a vehicle after HFMCD feeding (during the 21st and 22nd week). It has been reported that OCA treatment can ameliorate NASH and its complications in NASH animals [13,14,27]. The experimental groups thus consisted of Ctrl (n=7), NASH (n=10), Ctrl-OCA (n=7), and NASH-OCA (n=10) mice.

### Echocardiography

To determine the effects of chronic OCA treatment on NASH-associated cardiac dysfunction, cardiac functions were assessed by M-mode transthoracic echocardiography using a high-frequency and high-resolution echocardiography system (Vevo 3100, FUJIFILM VisualSonics Inc., Toronto, Canada) equipped with a 15- & 40- MHz ultrasound probe (Phillips Sonos 5500 System). To perform this procedure the mice were anaesthetised by inhalation of 2% isoflurane in oxygen. The images were collected in the short and long axes; the data represent the averaged values of 3~5 cardiac cycles. Diastolic and systolic volumes were acquired by applying Simpson's rule of discs to the serially acquired short axis images. Left ventricular diameters during diastole (LVIDd), left ventricular diameter during systole (LVIDs) and heart rate (HR) were determined from long axis M-modes. Cardiac output was determined by: SVxHR. Fractional shortening was calculated as: FS=(LVDd-LVDs)/LVDd×100% (LVDd, left ventricular diastolic dimension; LVDs, left ventricular systolic dimension). Relative wall thickness was

calculated as: (diastolic posterior wall thickness + diastolic anterior wall thickness)/ LVIDd. An experienced cardiologist and an echocardiography expert blinded to experimental groups performed all measurements.

## Body composition

Body composition (fat and lean mass) was measured three times in each mice using computed tomography. The results are presented as means of these measurements and expressed as a percentage of total body weight. After echocardiography and body composition measurements, mice in all four groups were sacrificed at 23nd week after either HFMCD diet or normal chow feeding.

## Serum and cardiac caspase-1 and IL-10 levels

Serum markers of myocardial injury, creatine kinase (CK) and cardiac troponin T, were measured using routine laboratory methods (SRL Inc., Tokyo, Japan). The cardiac tissue activity of caspase-1 was measured by cleavage of a fluorogenic substrate (Ac-YVAD-AMC) specific for caspase-1 (CaspACE, Promega, Madison, WI) whereas IL-10 levels were measured in heart homogenates by ELISA kits (abcam, Cambridge, MA, USA).

## Histopathologic examination

Liver and heart sections were stained with haematoxylin and eosin (H&E) for calculation of NAS scores and the degree of mononuclear cells infiltration was classified as mild (involvement < 25% of the tissue), moderate (25%–50% of the tissue), or severe (involvement > 50% of the tissue) [32]. The extent of heart (interstitial) fibrosis was visualized after Sirius Red staining for calculating collagen deposition in the total LV dimension (Sigma-Aldrich, St. Louis, MO, USA). Tissue abundance of CD3[+] T-cells and IL-10R[+] cells were quantified with confocal microscopy from 5-6 sections and 3-4 mice from each group.

## Intracellular level of FXR and inflammasome of primary isolated CD4[+]CD25[+] Treg cells

Mouse spleens from different groups were isolated using EasySep™ Mouse CD4[+]CD25[+] Treg cell isolation kit II/EasySep™ (StemCell Technologies, Vancouver, Canada). Then, the kit was used to isolate Treg cells from single-cell suspensions of splenocytes by positive selection to a purity of 95%, which was verified by flow cytometry. The isolated cells exhibited > 95% viability, confirmed by trypan blue dye exclusion. Unwanted cells were targeted for removal with biotinylated antibodies against non-T cells and streptavidin-coated magnetic particles. Labeled cells were separated using an EasySep™ magnet without using columns (manufacturer, city, country). All the culture media were supplemented with 10% fetal calf serum (BI), penicillin, streptomycin, glutamine, and 2-mercaptoethanol (Life technology, Foster City, CA, USA), and the CD4[+]CD25[+] Treg cells were cultured at 37˚C in 5% CO2.

## Percentage of proliferative and apoptotic cells among mice CD4[+]CD25[+] Treg cells

In addition to the intracellular levels of FXR, NLRP3 and caspase-1 by flow cytometry and immunofluorescence (IF) staining, percentage of proliferative (Ki-67[+]), and apoptotic (TUNEL[+] or annexin V[+]PI[+]) cells among first and second parts of CD4[+]CD25[+] Treg cells were assessed. Isotype control antibodies were used to determine the level of background staining and set up a specific signal threshold using a FACSCalibur flow cytometer (BD

Bioscience, Franklin Lakes, NJ, USA) and FlowJo software 7.6.1 (Tristar, Culver City, CA, USA). The maximal fluorescence intensity (MFI) of the isotype control was subtracted from the MFI of antibody-stained cells for each culture. Finally, the supernatant of the third part of CD4$^+$CD25$^+$ Treg cells were collected for using as condition medium for incubation of a rat embryonic cardiac cell line H9c2 cells in the following experiments.

## Effects of OCA on the interaction between substance release from Treg and H9c2 cells

The H9c2 cells were seeded into 24-well culture dishes at a density of 0.5×10$^5$ cells/well and propagated as myoblasts. At day 2 after seeding, the cultures were 80% confluent. Then, H9c2 cells were divided into six groups: 1.buffer; 2. HFMCDm; 3. Treg-NASHcm; 4.HFMCDm +Treg-Ctrl-cm; 5.HFMCDm+OCA (10μM); and 6. HFMCDm+OCA+siIL-10R groups. For HFMCDm group, H9c2 cells were exposed to HF-contained (200 μM palmitic acid) DMEM medium deficient in methionine and choline for 48 h. For Treg-NASHcm group, H9c2 cells were incubated with Treg-NASH [co-incubating with supernatant of Treg from NASH mice as condition medium (cm)]. In HFMCDm+Treg-Ctrl-cm group, H9c2 cells were co-incubated with HFMCDcm and Treg-Ctrl-cm [co-incubating with supernatant of Treg from Ctrl mice as condition medium (cm)]. The preliminary experiments reported that the dose of OCA (10μM) and siIL-10R (100μM) used in *in vitro* experiments did not have cytotoxic effects. So, in HFMCDm+OCA group, H9c2 cells were incubated with HFMCDm and 10μM of OCA. In HFMCDm+OCA+siIL-10R group, H9c2 cells were additional incubation with siIL-10R (100μM). Each group has six replicates. After various pretreatment, the percentage of proliferative (Ki-67$^+$), apoptotic (TUNEL$^+$ or annexin V$^+$PI$^+$) cells were assessed. For H9c2 differentiation marker evaluation, they were initially supplemented with 10% FBS (growth medium, GM) for 3 days and switched to the differentiation medium [DM; 1% FBS and 50 nM all trans-retinoic acid (RA, R2625; Sigma)] for 7 days. RA was added daily in the DM and were changed every other day among groups.

## Effects of pretreatment on H9c2 cell contraction

H9c2 cells contractions were measured by changes in planar surface area by culturing them in 6 mm flat bottom plates about 5 × 10$^5$ cells per plate. Percentage changes in planar surface areas of cells from baseline in response to treatment were observed using a video camera, at 12, 24, 48 hours, and compared with buffer group.

## Statistical analysis

Data are presented as the means and standard error of the samples analysed. Normal distribution test was performed prior to ANOVA test. Statistical significance was set at $p \leq 0.05$ with the nonparametric Kruskal–Wallis test and the Mann–Whitney $U$ test when appropriate with SPSS 19.0 (SPSS Inc., Chicago, IL, USA) and GraphPad Prism 6.0 (GraphPad Software, La Jolla, USA).

## Materials

FXR, IL-10R, IL-10, NLRP3, caspase-1, collagen-1, αSMA, MYH (myosin heavy chain kinase), myogenin and troponin-I (TNNI) antibodies were purchased from Cell Signaling Technology (Danvers, MA, USA), Abcam (Cambridge, MA, USA) and Santa Cruz (Dallas, Texas, USA).

**Table 1. Hemodynamic and cardiac injury parameters of mice in different groups.**

|  | Ctrl (=7) | NASH (n=10) | Ctrl-OCA (n=7) | NASH-OCA (n=10) |
|---|---|---|---|---|
| Mean arterial blood pressure (MAP, mmHg) | 118±6 | 129±7* | 112±4 | 121±5 |
| Heart rate (beats/min) | 388±29 | 390±36 | 385±41 | 420±45 |
| [cTnT, ng/mL] | 0.135±1.7* | 0.663±1.5† | 0.123±2.2 | 0.456±2.9# |
| [CK, IU/L] | 150±8 | 1517±134 | 1061±23 | 186±21 |

*P<0.05 vs. Ctrl group

#P<0.05 vs. NASH group.

## Results

### Systemic effects of chronic OCA treatment in NASH mice

In comparison with NASH mice, chronic OCA treatment normalised the lean mass and decreased fat mass, the hepatic NAFLD score, serum cardiac troponin T (cTnT) and CK, but did not change the heart rate and the mean blood pressure in the NASH-OCA mice (Table 1 and Fig 1A and 1B).

### Chronic OCA treatment reversed the cardiac dysfunction of NASH mice

Significantly, lower FS and higher LV mass were observed in the NASH group with cardiac dysfunctions compared to those in the Ctrl group (Fig 1B–1D). After chronic OCA treatment, the above-mentioned cardiac dysfunctions were normalised in NASH-OCA mice. Fig 1E and 1F shows that the percentage of NASH mice with severe cardiac inflammation (high number of mononuclear cell infiltration) was greater than that in the Ctrl group, whereas the percentage of NASH-OCA mice with severe cardiac inflammation was decreased after chronic OCA treatment.

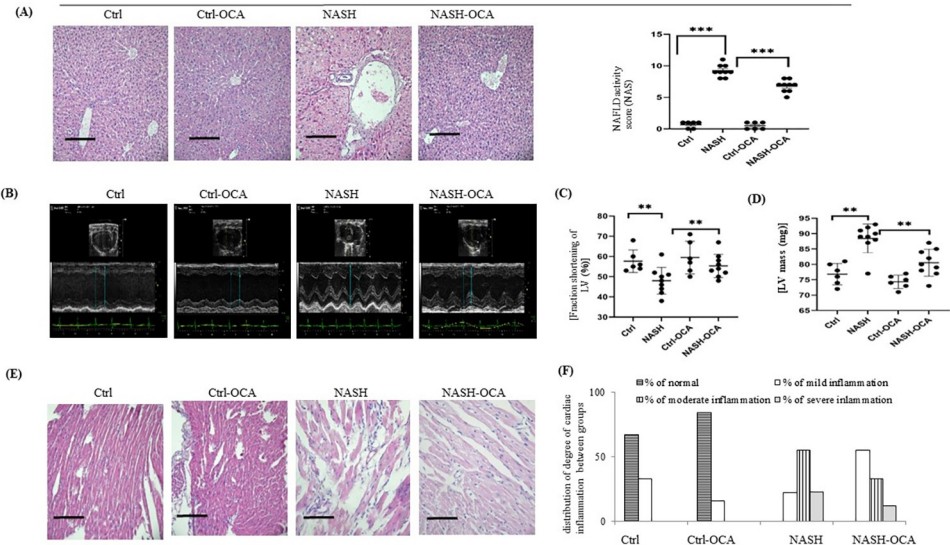

**Fig 1. Chronic OCA treatment suppresses NASH-related cardiac dysfunction. (A)** hepatic NAFLD score. **(B-C)** fraction of shortening (FS). **(D)** LV mass. **(E-F)** cardiac H&E staining for inflammation. 20x, Scale bar: 100μm; **,***: p<0.01 or <0.001 *vs*. Ctrl or NASH group. FS=(LVDd-LVDs)/LVDd×100%; LVDd: LV diastolic dimension; LVDs: lLV systolic dimension; LVEF=(LVEDV-LVESV)/LVEDV×100% (LVEDV, LV end-diastolic volume; LVESV, LV end-systolic volume). The degree of cardiac inflammation was classified as mild (mononuclear cell infiltration < 25% of the tissue), moderate (25%–50% of the tissue), or severe (> 50%).

## Cardiac FXR activation-related attenuation of cardiac inflammation was associated with less severe cardiac fibrosis and less hpo-contractility in NASH mice

The degree of cardiac fibrosis (% of sirius red-stained positive area) was more severe and suppressed by chronic OCA treatment in the NASH-OCA group than in the Ctrl group (Fig 2A). Lower cardiac FXR/SHP2/IL-10 and higher inflammasome (caspase-1) expression were associated with higher number of infiltrated CD3$^+$ and IL-10R$^+$ cells in the hearts of NASH mice (Fig 2A–2D). Significantly, chronic OCA treatment reversed the above-mentioned dysregulated markers in NASH mouse hearts (Fig 2A–2D). Fig 2E revealed that in NASH mice heart, severe cardiac inflammation was associated with high % of Sirius red stained positive area and low fraction shortening. In other words, severe cardiac inflammation was associated with severe cardiac fibrosis and cardiac hypo-contractility in NASH mice (Fig 2E).

## Severe inflammation/ fibrosis was associated with a decreased fraction of shortening in NASH mouse hearts

There were negative correlations between the percentage of OCA-decreased sirius red-stained positive area and the magnitude of the OCA-increased levels of FXR, IL-10R, and IL-10 (Fig 3A) protein expression in the hearts of mice in NASH group. Additionally, positive correlations were noted between the percentage of the OCA-increased FS and the magnitude of the OCA-increased levels of FXR, IL-10R, and IL-10 (Fig 3B).

## Chronic OCA treatment increased intracellular FXR in Treg cells from NASH mice splenocytes was associated with decreased inflammasome levels

In the primary isolated CD4$^+$CD25$^+$ Treg cells of the NASH mice, in comparison with the Treg cells from the Ctrl mice, lower intracellular FXR and higher inflammasome (NLRP3 and

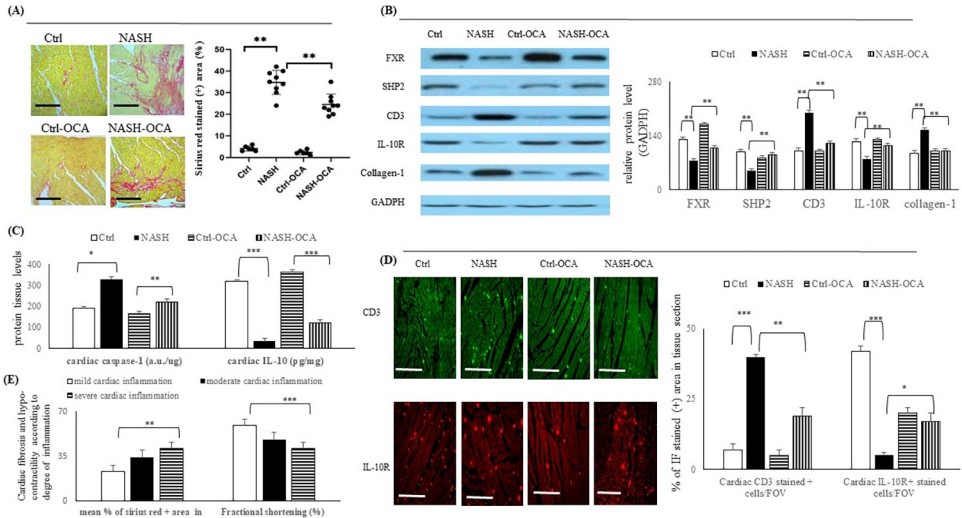

**Fig 2. Cardiac fibrosis of NASH mice was associated with increased infiltrated CD3$^+$ T cells but decreased cardiac Treg-related anti-inflammatory cytokines (IL-10/IL-10R).** (**A**) Sirius red-assessed cardiac fibrosis. (**B**) Various cardiac pathogenic protein markers expression. (**C**) Cardiac levels of caspase-1 and IL-10. (**D**) Mean degree of cardiac fibrosis and fraction of shortening among mice with mild, moderate, and severe cardiac inflammation. (**E**) Cardiac CD3$^+$T cells (green color) and IL-10R$^+$ Treg cells (red color) infiltration. **,***: p<0.01 or <0.001 *vs*. Ctrl or NASH group.

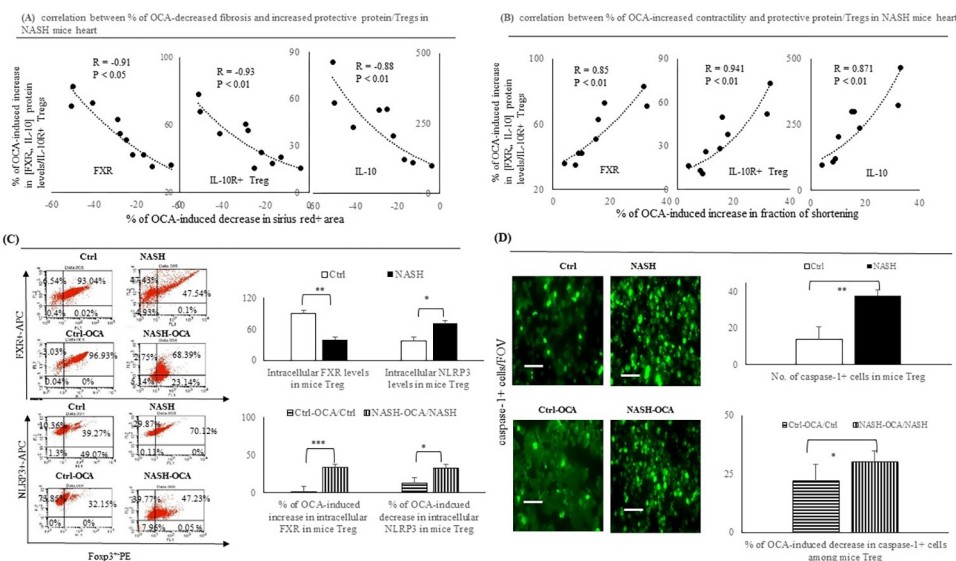

**Fig 3. OCA treatment-decrease in the collagen deposition was associated with the increase in the fraction of shortening in NASH hearts.** Correlation between (**A**) OCA-induced decrease in cardiac fibrosis (% of sirius red stained positive area) and (**B**) OCA-induced increase in fraction of shortening (FS) with OCA-induced increase in expressions of cardiac FXR/IL-10 levels, and infiltrated IL-10R+ Treg cells (CD4+CD25+) in NASH mice heart. (**C**) Flow cytometry assessed intracellular FXR and NLRP3 levels in Ctrl and NASH group as well as percentile change in FXR and NLRP3 levels by OCA treatment. (**D**) Immunofluorescence-measured caspase-1+ cells (green color) in primary isolated splenic regulatory T (Treg cells) of Crtl and NASH group as well as percentile decrease in caspase-1+ and IL-1β+ cells by OCA treatment. 20x, Scale bar: 100μm; **,***: p<0.01 or p<0.001 *vs*. Ctrl or NASH group.

caspase-1) levels were observed. In the primary Treg cells isolated from the NASH-OCA mice, OCA treatment upregulated intracellular FXR and downregulated inflammasomes (Fig 3C and 3D).

## Chronic OCA treatment increased the proliferative/active splenocytes and suppressed the apoptotic Treg cells in NASH-OCA mice

In comparison with the Ctrl group, primary Treg cells from NASH mice were characterised by a lower percentage of proliferative (Ki-67+) and active (HLADR+) cells, and a higher percentage of apoptotic cells, which were corrected by chronic OCA treatment (Fig 4A–4C). Significantly, the percentage of the OCA treatment-related suppression of apoptotic (TUNEL+ and Annexin-5+PI+) splenocytes and the increase in proliferative/active splenocytes were higher in the NASH-OCA group than those in the NASH and Ctrl groups, respectively.

## The effects of OCA on the HFMCD medium [HFMCDm]-activated inflammasomes and apoptosis in H9c2 cells were abolished by pre-treatment with siIL-10R

In comparison with buffer group, increased inflammasome (NLRP3 and caspase-1) levels, increased apoptotic [TUNEL+/Annexin-5+PI+] cells, and decreased proliferative [Ki-67+] cells were noted among the HFMCDm and Treg-NASHcm-treated H9c2 cells (Fig 5A–5D). Co-incubation with Treg-Ctrl-cm or OCA reversed the above-mentioned HFMCDm-induced changes in the H9c2 cells. The OCA-related attenuation of HFMCDm-induced changes in the H9c2 cell system was prevented by pre-treatment with siIL-10R.

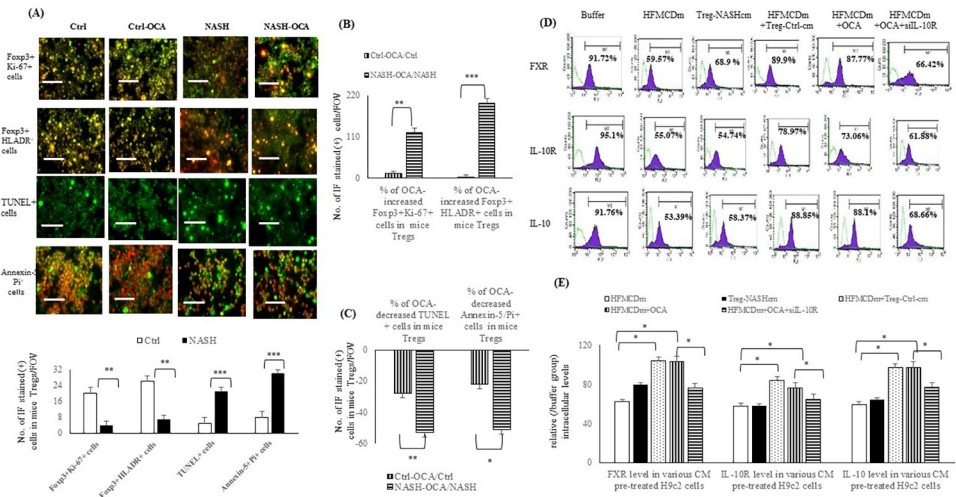

**Fig 4. OCA-treatment increases the splenic proliferative/activate Treg cells and decreased the apoptotic Treg cells among primary CD4⁺CD25⁺ splenocytes of NASH mice.** IF-based assessments of (**A**) Foxp3⁺Ki-67⁺ (orange color), (**B**) Foxp3⁺HLADR⁺ (orange color), (**C**) TUNEL⁺ (green color) (**D**) Annexin-5⁺Pi⁺ (green color) cells among primary CD4⁺CD25⁺ Treg cells from Ctrl, Ctrl-OCA, or NASH, NASH-OCA mice. (**E**) Foxp3⁺Ki-67⁺, Foxp3⁺HLADR⁺, TUNEL⁺, and Annexin-5⁺Pi⁺ cells among primary CD4⁺CD25⁺ Treg cells from Ctrl and NASH mice. (**F**) percentile increase in Foxp3⁺Ki-67⁺, Foxp3⁺HLADR⁺, and percentile decrease in TUNEL⁺, Annexin-5⁺Pi⁺ cells in primary CD4⁺CD25⁺ Treg cells by OCA treatment. **,***: $p<0.01$ or $p<0.001$ *vs.* Ctrl or NASH group.

## Acute OCA incubation inhibited the HFMCDm-related inhibition of the differentiation and contraction of H9c2 cells

Cardiac myosin heavy chain kinase and myogenin are markers of myoblast differentiation, whereas troponin I (TNNI) is an important marker for cardiac muscle contraction. Fig 6A and 6B reveal that HFMCDm and Treg-NASHcm significantly suppressed the levels of

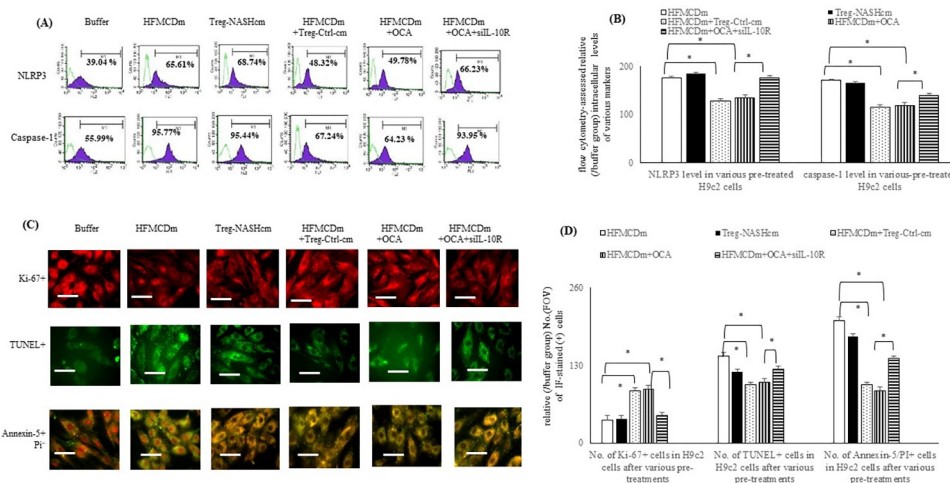

**Fig 5. Treg-NASHcm and HFMCDm increased the inflammasome and inhibited the proliferative H9c2 cells.** Flow cytometry-assessed relative intracellular levels (/buffer group) of (**A**) FXR, IL-10R, IL-10, (**B**) NLRP3, caspase-1 among various CM pre-treated H9c2. Bar graph of (**C**) immunofluorescence-assessed Ki-67⁺cells (red color), TUNEL⁺ cells (green color), and (**D**) Annexin-5⁺PI⁺ (orange color) cells in all H9c2 cells. Scale bars in all images are 100 mm; CD4⁺CD25⁺ Treg-NASH+OCA [supernatant of Treg cells from NASH mice as condition medium (cm) plus acute OCA incubation]; field of view (FOV).

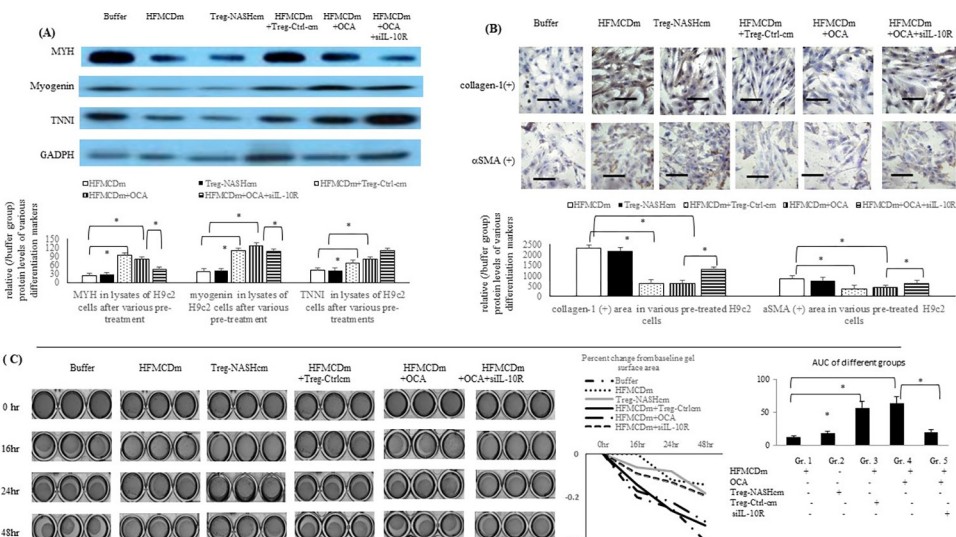

**Fig 6. Acute OCA incubation normalised the HFMCDm and Treg-NASHcm suppressed differentiation and contraction of H9c2 cells.** (**A**) Expression of the differentiation markers [cardiac myosin heavy chain (MYH), myogenin, TNNI] levels (/Buffer group) in supernatant of H9c2 cells. (**B**) The collagen-1 and αSMA positive among H9c2 cells. (**C**) Collagen gel contraction assay for evaluating the function of cardiomyocyte after various pretreatment. Scale bar = 20 μm. CD4+CD25+ Treg-NASH+OCA [supernatant of Treg cells from NASH mice as condition medium (cm) plus acute OCA incubation].

differentiation (MYH, myogenin, collagen-1, and αSMA) and contraction (TNNI) markers in the H9c2 cell lysates. Additionally, co-incubation with OCA or mice Treg-Ctrl-cm reversed the HFMCDm-suppressed differentiation markers. Significant suppression of the contraction of HFMCDm-treated H9c2 cells was observed, whereas OCA pre-treatment reversed the HFMCDm-inhibited contractility of H9c2 cells, which was corrected by pre-treatment with siIL-10R (Fig 6C).

## Discussion

NAFLD and heart failure are pathophysiological continuums of systemic inflammation that share common risk factors, comorbidities, and cardiac outcomes [32]. Cardiac inflammation and fibrosis are well-established cardiac dysfunction complications in patients with NASH [33,34]. It has been reported that activation of FXR attenuate cardiac inflammation and fibrosis in mice with diabetes and myocardial infarction [26,28]. Increased levels of inflammasomes including NLRP3 and caspase-1 play important pathogenic roles in cardiac inflammation, fibrosis, and impaired cardiac function [5–9]. Previous studies have revealed the protective effects of blocking the inflammasomes NLRP3 on cardiac dysfunction, such as reducing cardiac fibrosis and preserving contractile function in mice and in patients with cardiac dysfunction [8,9,35,36]. In obese and cholestatic mice, activation of FXR is associated with significant suppression of inflammasome-related pathogenesis [11,14]. Chronic OCA administration inhibited the LV hypertrophy and normalised FS and LVEF by inhibition of inflammasome in diabetic mice with dilated cardiomyopathy [27]. In the current study, the cardioprotective effects of chronic FXR activation with OCA were firstly reported in NASH mice with cardiac dysfunction (Fig 7).

Functional regulatory T cells typically express IL-10/IL-10R, which has anti-inflammatory activity [37]. Among the Treg cells of the Ctrl mice in our study, normal FXR and IL-10/IL10R signals, normal inflammasome levels, normal frequency of proliferative/activated cells, and

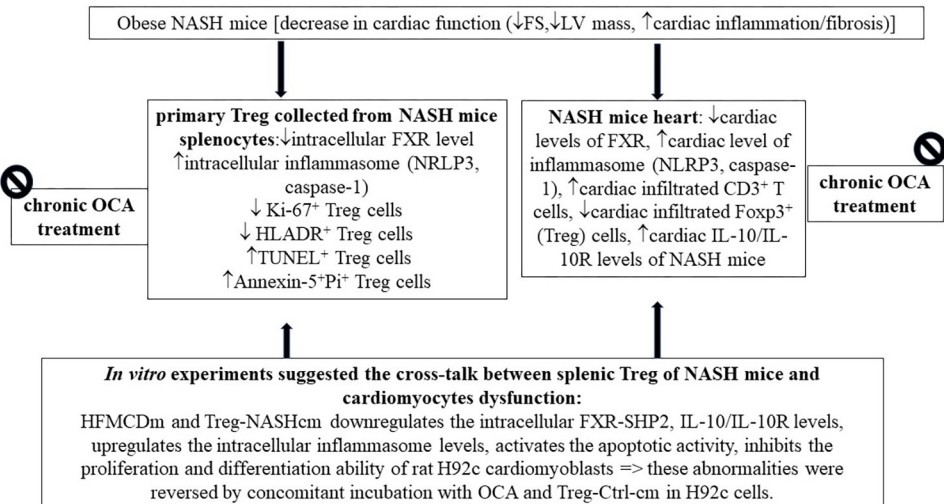

**Fig 7. Graphical summary of the pathogenic mechanisms and effects of chronic OCA treatments on mice with NASH-related cardiac dysfunction.** FM: Fat mass; FS: Fractional shortening; FXR: Farnesoid X receptor; SHP2: Small heterodimer partner 2; Treg: T regulatory cell; Treg-NASH: Treg cells of NASH mice; cm: Condition medium; HFMCD: High-fat and methionine and choline deficiency.

low apoptotic cells were associated with normal heart function and histology compared to those in the NASH group. A reduction in the frequency of Tregs has been reported in patients with chronic heart failure, and dilated cardiomyopathy [17,18]. Activated inflammasomes can negatively regulate the number of circulating or local Treg subsets, their function, and its cardiac protective effects [20]. Recent study reported that. T lymphocyte depletion ameliorates age-related metabolic impairments in mice [38]. The study also reported that age-related insulin resistance (IR) was associated with increased in T cells infiltration in skeletal muscle of old mice with IR [38]. Previous study reported a close relationship between IR and disease progression and complications development of NAFLD [39]. The correlation between cardiac insulin resistance and progression of cardiac dysfunction had been reported [40]. Accordingly, the pathogenic links between insulin resistance (IR) and T cell dysregulation-mediated NASH-associated cardiac dysfunction should be elucidated in future study.

Cardiomyocyte differentiation and proliferation are important to avoid cell apoptosis to preserve heart function after ischemic injury. Tregs-related signals such as Foxp3 and IL-10/IL10R can directly facilitate cardiomyocyte differentiation and proliferation to maintain post-ischemic cardiac function [41–43]. Expansion of Tregs ameliorates adverse post-injury cardiac remodelling [44]. Both *in vivo* and *in vitro* experiments in the current study revealed that acute and chronic OCA treatment upregulated Treg (Foxp3 and IL-10/IL10R) signals and maintaining the proliferation and contraction in cardiomyocytes in NASH mice heart.

Typically, HFD-induced NASH rats are characterized by the elevated serum markers of cardiac injury, including CK and troponin T [45]. Circulating cardiac troponin T is the preferred biomarker of cardiac injuries following necrosis and ischaemic stress [46]. In our *in vitro* study, the attenuation of the HFMCD medium-suppressed proliferation, differentiation, and contraction of cardiomyocytes by concomitant incubation of Treg supernatant from the Ctrl group [condition medium] or OCA were accompanied by the *in vivo* suppression of circulating cardiac injury markers, including CK and troponin T in OCA treatment-related NASH mice.

Activated inflammasomes can negatively regulate IL-10/IL10R signals and have corresponding cardiac damaging effects [22–24]. By binding with IL-10R, Treg-represented

cytokine IL-10 inhibit the synthesis and activation of inflammasome including NLRP3 and pro-caspase-1 [47,48]. Evidence suggests that inflammasomes exerts pro-inflammation, pro-apoptosis, and anti-contraction effects on cardiomyocytes [49]. Therefore, in our study, it is reasonable to find that the OCA-treatment-related upregulation of IL-10 and inhibition of IL-1β suppressed cardiac apoptotic activity and increased FS in the NASH mice.

Chronic OCA treatment can sustainedly reduce liver stiffness and fibrosis of NASH patients with advanced hepatic fibrosis [50]. Although some NASH patients developed mild to moderate pruritus early after OCA therapy initiation, the general quality of life was improved [12,51,52]. The activity/energy domains of chronic liver disease questionnaire of NASH patients (CLDQ-NASH) addresses symptoms of NASH patients are significant improved after OCA treatment [52]. Myocardial remodelling and dysfunction have been reported in NAFLD patients and are correlated with disease progression [3,4]. In diabetic mice, FXR activation by OCA treatment improves myocardial dysfunction [26–28]. In the current study, the cardio-protective effects of chronic FXR activation with OCA were firstly reported in NASH mice with cardiac dysfunction. From the initial positive results of this report, the effects of chronic OCA treatment need to evaluate in NASH patients in future study.

Previous studies reported that NAFLD is a common disease in the elderly, and with aging, the liver undergoes substantial changes in structure and function that are associated with significant impairment of many hepatic metabolic and detoxification activities [53,54]. So, the limitation of our current study is lack of the exploration of the impact of structural and function changes of liver on the NASH-associated cardiac function in aged mice of our study. In other words, future study needs to include the examine the interaction between liver and heart in aged NASH mice.

In mice with NASH-related cardiac dysfunction, through *in vivo* and *in vitro* approaches, this study revealed the pathogenic mechanisms, including downregulation of FXR and Treg-represented (IL-10/IL10R) signals, upregulation of inflammasomes, increase in the cardiac CD3[+] T cell infiltration, increased crosstalk between dysregulated cardiac Treg cells and cardiomyocytes, increased apoptosis, and reduced proliferation, differentiation, and contraction of cardiomyocytes. In current study, the degree of downregulation of the FXR and Treg-represented (IL-10/IL10R) signals were shown to be negatively related to the increase in NASH-related cardiac fibrosis and systolic dysfunction in NASH mice. *In vitro* experiments using primary Treg cells and a H9c2 cardiomyocyte cell line revealed that pre-treatment with siRNA of IL-10R (siIL-10R) abolished the FXR agonist OCA-related acute effects on HFMCD medium-activated pathogenic markers. Through multimodalities-based approaches, the current study observed that activation of FXR by chronic OCA treatment normalised cardiac Treg-represented (IL-10/IL10R) signals and attenuated the abovementioned pathogenic markers in NASH mouse hearts. Overall, this study revealed the downregulation of FXR and IL-10/IL-10R-related serological, functional and pathogenic signalling mechanisms, and phenotypic abnormalities of NASH-related cardiomyopathy at systemic and cellular levels in a mouse model.

In conclusion, chronic FXR activation with OCA is a potential strategy to inhibit cardiac inflammation, cardiac fibrosis, cardiac apoptosis, and cardiac hypocontractility, restore the cardiac regulatory T cell subset and function, and subsequently improve cardiac dysfunction in NASH mice.

## Supporting information

**S1 Raw data.**
(XLSX)

**S1 Raw images.**
(PDF)

## Acknowledgments

We would like to thank all staffs involved and supported grants in this study.

## Author Contributions

**Data curation:** Han-Chieh Lin.

**Formal analysis:** Chia-Chang Huang, Han-Chieh Lin.

**Funding acquisition:** Chia-Chang Huang, Ying-Ying Yang, Tzung-Yan Lee, Ming-Chih Hou.

**Investigation:** Szu-Yu Liu, Han-Chieh Lin.

**Methodology:** Szu-Yu Liu, Ying-Ying Yang, Shiang-Fen Huang, Han-Chieh Lin.

**Project administration:** Szu-Yu Liu, Chia-Chang Huang, Ying-Ying Yang, Shiang-Fen Huang, Tzung-Yan Lee.

**Resources:** Ying-Ying Yang, Shiang-Fen Huang, Ming-Chih Hou.

**Software:** Ying-Ying Yang, Shiang-Fen Huang, Tzu-Hao Li, Ming-Chih Hou.

**Supervision:** Ying-Ying Yang, Tzung-Yan Lee, Tzu-Hao Li, Ming-Chih Hou.

**Validation:** Ying-Ying Yang, Tzu-Hao Li.

**Visualization:** Tzung-Yan Lee.

**Writing – original draft:** Szu-Yu Liu, Ying-Ying Yang, Tzu-Hao Li.

**Writing – review & editing:** Ying-Ying Yang.

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
