## [Decision Letter · Decision Letter 0]

30 Aug 2022

PONE-D-22-19785Obeticholic acid treatment ameliorates the cardiac dysfunction in NASH micePLOS ONE

Dear Dr. Yang,

Thank you for submitting your manuscript to PLOS ONE. After careful consideration, we feel that it has merit but does not fully meet PLOS ONE’s publication criteria as it currently stands. Therefore, we invite you to submit a revised version of the manuscript that addresses the points raised during the review process.

ACADEMIC EDITOR: All issues raised by expert reviewers are required.

We look forward to receiving your revised manuscript.

Kind regards,

Vincenzo Lionetti, M.D., PhD

Academic Editor

PLOS ONE

Journal Requirements:

2.We suggest you thoroughly copyedit your manuscript for language usage, spelling, and grammar. If you do not know anyone who can help you do this, you may wish to consider employing a professional scientific editing service. 

"This work was supported in part by MOST-110-2634-F-A49-005, MOST-109-2314-B-010-032-MY3 and MOST-110-2511-H-A491-504-MY3 from the National Science Council (Y.Y.Y. and H.C.L.), and 111EA-009, V111EA-010, V111C-018, V111C-038, VTA111-A-4-3 (Y.Y.Y. and H.C.L.) by the Taipei Veterans General Hospital."

Please respond by return e-mail so that we can amend your financial disclosure and competing interests on your behalf.

"This work was supported in part by MOST-110-2634-F-A49-005, MOST-109-2314-B-010-032-MY3 and MOST-110-2511-H-A491-504-MY3 from the National Science Council (Y.Y.Y. and H.C.L.), and 111EA-009, V111EA-010, V111C-018, V111C-038, VTA111-A-4-3 (Y.Y.Y. and H.C.L.) by the Taipei Veterans General Hospital."

"This work was supported in part by MOST-110-2634-F-A49-005, MOST-109-2314-B-010-032-MY3 and MOST-110-2511-H-A491-504-MY3 from the National Science Council (Y.Y.Y. and H.C.L.), and 111EA-009, V111EA-010, V111C-018, V111C-038, VTA111-A-4-3 (Y.Y.Y. and H.C.L.) by the Taipei Veterans General Hospital."

Reviewers' comments:

Reviewer's Responses to Questions

**Comments to the Author**

1. Is the manuscript technically sound, and do the data support the conclusions?

Reviewer #1: Partly

Reviewer #2: Yes

2. Has the statistical analysis been performed appropriately and rigorously? 

Reviewer #1: Yes

Reviewer #2: Yes

3. Have the authors made all data underlying the findings in their manuscript fully available?

Reviewer #1: Yes

Reviewer #2: Yes

4. Is the manuscript presented in an intelligible fashion and written in standard English?

Reviewer #1: No

Reviewer #2: No

5. Review Comments to the Author

Reviewer #1: In this investigation, the authors attempt to evaluate the in vivo and in vitro effects and mechanisms of 2-weeks of Foresaid X receptor (FXR) agonist, obeticholic acid (OCA), treatment on the Treg dysregulation-related cardiac dysfunction in NASH mice (NASH-OCA) at systemic, tissue and cellular levels. This study is interesting and timely as it suggests that chronic FXR activation with OCA has the potential to activate IL-10/IL-10R signaling, thus reversing the cardiac regulatory T cells dysfunction, and improving NASH-related cardiomyopathy.

- The reviewer identified the following concerns:

-Nonalcoholic fatty liver disease (NAFLD) is a common disease in the elderly, and with aging, the liver undergoes substantial changes in structure and function that are associated with significant impairment of many hepatic metabolic and detoxification activities (PMID: 34378088, PMID: 34117600). With that bein said the presented studies examine the myocardial remodeling effects of a diet that models NAFLD and NASH on 8 month old mice, the authors are encouraged to cite this work and include a description of this limitation for completeness

-How do the authors interpret their finding in light of recent evidence suggesting that T lymphocyte depletion ameliorates age-related metabolic impairments in mice.(PMID: 33893902 )

-Numerous spelling/syntax errors compromise the readability and understanding of many passages, thus preventing an effective and complete review of this work. For example (The dysregulated signals between cardiomyocytes and regulatory T

(Treg) cells are involve in the development of nonalcoholic steatohepatitis)

- "10mg/kg/day, using the azert osmotic pump". I believe the authors intend to say the alzet osmotic minipump, please cite the company appropriately per journal guidelines.

- The figures need extensive reformatting, the histology figures are all out of alignment and the scale bars are placed seemingly randomly, with some ending outside of the image. Colors in immunohistology images are not explained in the legend. Some bar graphs lack statistics and error bars.

Reviewer #2: In this paper, the Authors explored the interplay between non-alcoholic steatohepatitis (NASH) and cardiac dysfunction using both in vivo and in vitro experiments. The Authors confirm that NASH can induce cardiac dysfunction mainly through the dysregulation of regulatory T cell activity, and the consequent activation of proinflammatory pathways. Reduced signalling of the farnesoid X receptor seems to play a pivotal role in this process, which can be reversed by the administration of an agonist, namely obeticholic acid (OCA). The study design is straightforward and well-articulated; the methods are consistent. The main issue with is paper is the English language, which requires extensive revision starting from the Abstract. Even the name of the main receptor studied in this paper is written incorrectly (“Foresaid X receptor” instead of “Farnesoid X receptor”)! Besides that, here are some further comments:

• At which time point were mice sacrificed? From what is reported in the Methods, NASH-OCA mice received a 2-week treatment with OCA after completion of the NASH induction protocol: were these mice sacrificed 2 weeks later than control and NASH (without OCA) ones?

• Please discuss in more detail the potential clinical implications of the findings of this paper in the Discussion. Notably, OCA has been already studied in humans for the treatment of NASH, but the randomized controlled trial investigating the drug (FLINT trial; doi.org/10.1016/S0140-6736(14)61933-4) was stopped prematurely due to long-term safety concerns regarding lipid abnormalities occurring in the intervention arm.

• The Abstract lacks a real “aim”.

• There is no “Conclusions” section at the end of the paper.

6. PLOS authors have the option to publish the peer review history of their article (what does this mean?). If published, this will include your full peer review and any attached files.

Reviewer #1: No

Reviewer #2: No

---

## [Author Response · Author response to Decision Letter 0]

17 Sep 2022

PONE-D-22-19785

Obeticholic acid treatment ameliorates the cardiac dysfunction in NASH mice

Response to the comments of associated editors: 

comments:

Thank you for submitting your manuscript to PLOS ONE. After careful consideration, we feel that it has merit but does not fully meet PLOS ONE’s publication criteria as it currently stands. Therefore, we invite you to submit a revised version of the manuscript that addresses the points raised during the review process. All issues raised by expert reviewers are required.

Response: Thanks for giving us this opportunity to revise our manuscript. Meanwhile, we are very appreciated for your and reviewer recommendation to improve the background of our manuscript. The point-to-point response had been included, the whole manuscript had been gone through and editing by native English speaker. The original images of western blots had been included as supporting information. 

Reviewer #1: In this investigation, the authors attempt to evaluate the in vivo and in vitro effects and mechanisms of 2-weeks of Foresaid X receptor (FXR) agonist, obeticholic acid (OCA), treatment on the Treg dysregulation-related cardiac dysfunction in NASH mice (NASH-OCA) at systemic, tissue and cellular levels. 

Comment 1 :This study is interesting and timely as it suggests that chronic FXR activation with OCA has the potential to activate IL-10/IL-10R signaling, thus reversing the cardiac regulatory T cells dysfunction, and improving NASH-related cardiomyopathy.

Comment 1 : thanks for your positive comments about this manuscript and your very constructive comments are responded point-to-point in the following paragraphs.

- The reviewer identified the following concerns:

Comment 2 :-Nonalcoholic fatty liver disease (NAFLD) is a common disease in the elderly, and with aging, the liver undergoes substantial changes in structure and function that are associated with significant impairment of many hepatic metabolic and detoxification activities (PMID: 34378088, PMID: 34117600). With that bein said the presented studies examine the myocardial remodeling effects of a diet that models NAFLD and NASH on 8-month-old mice, the authors are encouraged to cite this work and include a description of this limitation for completeness

Response 2: Thanks for these two important manuscripts that your provided. Really, our study are focusing on examine the myocardial remodeling effects of diet-induced NASH model on 8 month old mice. From recent studies that your provided, nonalcoholic fatty liver disease (NAFLD) is a common disease in the elderly, and with aging, the liver undergoes substantial changes in structure and function that are associated with significant impairment of many hepatic metabolic and detoxification activities [GeroScience 43,1975–1993 [PMID: 34117600]; GeroScience , 44:835–845 [PMID: 34378088]]. So, the limitation of our current study is lack of the exploration of the impact of structural and function changes of liver on the NASH-associated cardiac function in mice. In other words, future study needs to include the examine the interaction between liver and heart in old age NASH mice. These discussions had been included in “discussion” section [page 13, paragraph 2]. 

-Comment 3 :How do the authors interpret their finding in light of recent evidence suggesting that T lymphocyte depletion ameliorates age-related metabolic impairments in mice.(PMID: 33893902)

Response 3: thanks for providing this important new reference, we had incorporated the new report [GeroScience 2021; 43,1331–1347. PMID: 33893902] with the findings of our current study in “revised” version. This reference [GeroScience 2021; 43,1331–1347. PMID: 33893902] reported that T lymphocyte depletion ameliorates age-related metabolic impairments in mice. In that study age-related insulin resistance (IR) was associated with increased in T cells infiltration in skeletal muscle of old mice with IR [PMID: 33893902]. Previous study reported a close relationship between IR and disease progression, and development of NAFLD-related complications [Hepatology 2019;70(2),711-724]. The correlation between cardiac insulin resistance and progression of cardiac dysfunction had been reported [Int J Mol Sci. 2019; 20(14), 3552]. Accordingly, the pathogenic links between insulin resistance (IR) and T cell dysregulation-mediated NASH-associated cardiac dysfunction should be elucidated in future study. The discussion and new references [ref. 38-40] had been included in “discussion” section of our study [page 11,paragraph 3, line 8-9;page 12, paragraph 1]. 

Comment 4: Numerous spelling/syntax errors compromise the readability and understanding of many passages, thus preventing an effective and complete review of this work. For example (The dysregulated signals between cardiomyocytes and regulatory T (Treg) cells are involve in the development of nonalcoholic steatohepatitis)

Response 4: Thanks for your comments, we had checked the possible spelling/syntax errors throughout the whole manuscript, Figures/legends, Tables and Figures, repeatedly. The area of changes had been highlighted in yellow. 

Comment 5: "10mg/kg/day, using the azert osmotic pump". I believe the authors intend to say the alzet osmotic minipump, please cite the company appropriately per journal guidelines.

Response 5: Thanks for your very comment about the wrong spelling of Alzet osmotic pump. In revised version, the alzet ® osmotic pump (DURECT, Cupertino, CA) had been corrected in “method” section [page 4, paragraph 2, line 16]. 

Comment 6: The figures need extensive reformatting, the histology figures are all out of alignment and the scale bars are placed seemingly randomly, with some ending outside of the image. Colors in immunohistology images are not explained in the legend. Some bar graphs lack statistics and error bars.

Response 6: thanks for your comments about the histology figures, we had carefully adjusted the alignment and the scale bars of all histology figures. Meanwhile, the colors (green or red color) in immunohistology images (Figure 2D, 3D, 4A,5C). Further, the bar graphs lack of statistics and error bars had been included.

 

Reviewer #2: In this paper, the Authors explored the interplay between non-alcoholic steatohepatitis (NASH) and cardiac dysfunction using both in vivo and in vitro experiments. The Authors confirm that NASH can induce cardiac dysfunction mainly through the dysregulation of regulatory T cell activity, and the consequent activation of proinflammatory pathways. Reduced signalling of the farnesoid X receptor seems to play a pivotal role in this process, which can be reversed by the administration of an agonist, namely obeticholic acid (OCA). 

Comment 1: The study design is straightforward and well-articulated; the methods are consistent.

Response 1: thanks for your positive comments about this manuscript and your very constructive comments are responded point-to-point in the following paragraphs.

Comment 2: The main issue with is paper is the English language, which requires extensive revision starting from the Abstract. Even the name of the main receptor studied in this paper is written incorrectly (“Foresaid X receptor” instead of “Farnesoid X receptor”)! 

Response 2: thanks for your comments about English language, we had extensively revised the whole manuscript. The certification for English editing had been included as supplement files. The area of changes had been highlighted in yellow color. The “Foresaid X receptor” had been corrected.

Comment 3: Besides that, here are some further comments:• At which time point were mice sacrificed? From what is reported in the Methods, NASH-OCA mice received a 2-week treatment with OCA after completion of the NASH induction protocol: were these mice sacrificed 2 weeks later than control and NASH (without OCA) ones?

Response 3: Thanks for giving us this opportunity to clarify these points. After echocardiography and body composition measurements, mice in both NASH and NASH-OCA groups were sacrificed at 23nd week after either diet (HFMCD) diet or normal chow feeding. This description had been included in revised method [page 5, paragraph 3].

Comment 4:• Please discuss in more detail the potential clinical implications of the findings of this paper in the Discussion. Notably, OCA has been already studied in humans for the treatment of NASH, but the randomized controlled trial investigating the drug (FLINT trial; doi.org/10.1016/S0140-6736(14)61933-4) was stopped prematurely due to long-term safety concerns regarding lipid abnormalities occurring in the intervention arm.

Response 4: Thanks for your comments about the inclusion of the potential clinical implications of the findings of this paper in the Discussion. According to your comments, the following discussion had been included in “revised” version [page 13, paragraph 3]. Chronic treatment with OCA in NASH patients with fibrosis stage F2 or F3 can sustainedly reduce liver stiffness and fibrosis [J Hepatol 2022;76(3), 536-548]. Although some of these NASH patients developed mild to moderate pruritus, the general quality of life was improved [Lancet, 2015;385(9972):956-65;Lancet 2019;394 (10215), 2184-2196]. In general, mild pruritus occurs early after OCA therapy initiation and does not worsen over time [Clin Gastroenterol Hepatol 2022; 20(9),2050-2058.e12]. Especially, the activity/energy domains of chronic liver disease questionnaire of NASH patients (CLDQ-NASH) addresses symptoms of NASH patients were significant improved after OCA treatment [Clin Gastroenterol Hepatol 2022; 20(9),2050-2058.e12]. Myocardial remodelling and dysfunction have been reported in NAFLD patients and are correlated with disease progression [J Hepatol 2013;58(4), 757-762; Nat Rev Gastroenterol Hepatol 2018;15(7), 425-439]. Increased left ventricle hypertrophy and ventricular diastolic dysfunction have been reported in NAFLD patients [Rev Gastroenterol Hepatol 2018;15(7), 425-439]. In diabetic mice, FXR activation by OCA treatment improves myocardial dysfunction by inhibiting cardiac inflammation and fibrosis [Curr Protein Pept Sci 2019;20(10), 976-983; Cardiovas Res 2019;114(10),1335-1349]. In the current study, the cardioprotective effects of chronic FXR activation with OCA were firstly reported in NASH mice with cardiac dysfunction. From the initial positive results of this report, the effects of chronic OCA treatment need to evaluate in NASH patients in future study. This description had been included in revised method [page 12, paragraph 4; page 13, paragraph 1]. 

•Comment 5: The Abstract lacks a real “aim”.

Response 5: Thanks for your comments about the “real aim” in the abstract section. In revised version, the “aim” has been included in abstract as below “This study aimed to evaluate the mechanisms and effects of chronic treatment with the FXR agonist OCA on cardiac Treg cell dysfunction-related, inflammasome-mediated, and NASH-associated cardiac dysfunction in mice” [page 2].

Comment 6: There is no “Conclusions” section at the end of the paper.

Response 6: Thanks for your comments about the “conclusion” section. In “revised” version, the “Conclusions” section had been included at the end of the paper [page 13, paragraph 4]. “In conclusion, chronic FXR activation with OCA is a potential strategy to inhibit cardiac inflammation, cardiac fibrosis, cardiac apoptosis, and cardiac hypocontractility, restore the cardiac regulatory T cell subset and function, and subsequently improve cardiac dysfunction in NASH mice.”.

---

## [Decision Letter · Decision Letter 1]

5 Oct 2022

PONE-D-22-19785R1Obeticholic acid treatment ameliorates the cardiac dysfunction in NASH micePLOS ONE

Dear Dr. Yang,

Thank you for submitting your manuscript to PLOS ONE. After careful consideration, we feel that it has merit but does not fully meet PLOS ONE’s publication criteria as it currently stands. Therefore, we invite you to submit a revised version of the manuscript that addresses the points raised during the review process.

Acronyms should be written in extended form the first time they are reported (for example, what "FXR" stands for is not explained in the revised Abstract.

We look forward to receiving your revised manuscript.

Kind regards,

Vincenzo Lionetti, M.D., PhD

Academic Editor

PLOS ONE

Journal Requirements:

Reviewers' comments:

Reviewer's Responses to Questions

**Comments to the Author**

1. If the authors have adequately addressed your comments raised in a previous round of review and you feel that this manuscript is now acceptable for publication, you may indicate that here to bypass the “Comments to the Author” section, enter your conflict of interest statement in the “Confidential to Editor” section, and submit your "Accept" recommendation.

Reviewer #1: All comments have been addressed

Reviewer #2: All comments have been addressed

2. Is the manuscript technically sound, and do the data support the conclusions?

Reviewer #1: Yes

Reviewer #2: Yes

3. Has the statistical analysis been performed appropriately and rigorously? 

Reviewer #1: Yes

Reviewer #2: Yes

4. Have the authors made all data underlying the findings in their manuscript fully available?

Reviewer #1: Yes

Reviewer #2: Yes

5. Is the manuscript presented in an intelligible fashion and written in standard English?

Reviewer #1: Yes

Reviewer #2: Yes

6. Review Comments to the Author

Reviewer #1: The authors have addressed the reviewer's comments and the manuscript has been substantially improved

Reviewer #2: The Authors have revised and improved the manuscript following the Reviewers’ comments.

I have just one further minor suggestion:

- Acronyms should be written in extended form the first time they are reported (for example, what "FXR" stands for is not explained in the revised Abstract)

7. PLOS authors have the option to publish the peer review history of their article (what does this mean?). If published, this will include your full peer review and any attached files.

Reviewer #1: No

Reviewer #2: No

---

## [Author Response · Author response to Decision Letter 1]

7 Oct 2022

PONE-D-22-19785R1

Obeticholic acid treatment ameliorates the cardiac dysfunction in NASH mice

PLOS ONE

Response to editor office comments:

Thank you for submitting your manuscript to PLOS ONE. After careful consideration, we feel that it has merit but does not fully meet PLOS ONE’s publication criteria as it currently stands. Therefore, we invite you to submit a revised version of the manuscript that addresses the points raised during the review process.

-Comment 1: Acronyms should be written in extended form the first time they are reported (for example, what "FXR" stands for is not explained in the revised Abstract.

Response 1: Thanks for editor broad team for giving us this opportunity to revise this work. In “revised” abstract and whole manuscript, the full name of acronyms that appearing first time had been included. 

Reviewers' comments:

Comments to the Author

Reviewer #1 comments: The authors have addressed the reviewer's comments and the manuscript has been substantially improved.

Response to reviewer 1 comment: Thanks for your very constructive comments to improve our manuscript. 

Reviewer #2 comments:The Authors have revised and improved the manuscript following the Reviewers’ comments. I have just one further minor suggestion:

Comment 1- Acronyms should be written in extended form the first time they are reported (for example, what "FXR" stands for is not explained in the revised Abstract)

Response 1: Thanks for editor broad team for giving us this opportunity to revise this work. In “revised” abstract and whole manuscript, the full name of acronyms that appearing first time had been included.

---

## [Editor Report · Decision Letter 2]

12 Oct 2022

Obeticholic acid treatment ameliorates the cardiac dysfunction in NASH mice

PONE-D-22-19785R2

Dear Dr. Yang,

We’re pleased to inform you that your manuscript has been judged scientifically suitable for publication and will be formally accepted for publication once it meets all outstanding technical requirements.

Kind regards,

Vincenzo Lionetti, M.D., PhD

Academic Editor

PLOS ONE
---

## [Editor Report · Acceptance letter]

9 Nov 2022

PONE-D-22-19785R2 

Obeticholic acid treatment ameliorates the cardiac dysfunction in NASH mice 

Dear Dr. Yang:

I'm pleased to inform you that your manuscript has been deemed suitable for publication in PLOS ONE. Congratulations! Your manuscript is now with our production department. 

Kind regards, 

on behalf of

Prof. Vincenzo Lionetti 

Academic Editor

PLOS ONE